

**Sensitivity of modeled Indian Monsoon to Chinese and Indian aerosol**
**emissions**
Peter Sherman[1], Meng Gao[2,3], Shaojie Song[3], Alex T. Archibald[4,5], Nathan Luke Abraham[4,5],
Jean-François Lamarque[6], Drew Shindell[7], Gregory Faluvegi[8,9], Michael B. McElroy[1,3]
[1]Department of Earth and Planetary Sciences, Harvard University, Cambridge, Massachusetts,
United States
[2]Department of Geography, Hong Kong Baptist University, Hong Kong SAR, China
[3]School of Engineering and Applied Sciences, Harvard University, Cambridge, Massachusetts,
United States
[4]National Centre for Atmospheric Science, University of Cambridge, Cambridge, UK
[5]Department of Chemistry, University of Cambridge, Cambridge, UK
[6]National Center for Atmospheric Research, Boulder, Colorado, USA
[7]Nicholas School of the Environment, Duke University, Durham, NC, USA
[8]NASA Goddard Institute for Space Studies, New York, NY
[9]Center for Climate Systems Research, Earth Institute, Columbia University, New York NY


## Abstract

The South Asian summer monsoon supplies over 80% of India's precipitation. Industrialization
over the past few decades has resulted in severe aerosol pollution in India. Understanding
monsoonal sensitivity to aerosol emissions in general circulation models (GCMs) could improve
predictability of observed future precipitation changes. The aims here are (1) to assess the role of
aerosols on India's monsoon precipitation and (2) to determine the roles of local and regional
emissions. For (1), we study the Precipitation Driver Response Model Intercomparison Project
experiments. We find that the precipitation response to changes in black carbon is highly uncertain
with a large intermodel spread due in part to model differences in simulating changes in cloud
vertical profiles. Effects from sulfate are clearer; increased sulfate reduces Indian precipitation, a
consistency through all of the models studied here. For (2), we study bespoke simulations, with
reduced Chinese and/or Indian emissions in three GCMs. A significant increase in precipitation
(up to ~20%) is found only when both countries' sulfur emissions are regulated, which has been
driven in large part by dynamic shifts in the location of convective regions in India. These changes
have the potential to restore a portion of the precipitation losses induced by sulfate forcing over
the last few decades.



## 34     **Significance Statement**

The aims here are to assess the role of aerosols on India's monsoon precipitation and to determine
the relative contributions from Chinese and Indian emissions using CMIP6 models. We find that
increased sulfur emissions reduce precipitation, which is primarily dynamically driven due to
spatial shifts in convection over the region. A significant increase in precipitation (up to ~20%) is
found only when both Indian and Chinese sulfate emissions are regulated.


## 1. Introduction

The South Asian summer monsoon is the dominant weather pattern over India, lasting typically from June to September. Over this period, southwesterly winds transport warm, moist air from the Arabian Sea onto the Indian subcontinent, supplying roughly 80% of the region's annual rainfall (Turner and Annamalai, 2012). Since the monsoon provides such a significant source for India's water supply, changes in its strength and position would have important socioeconomic implications including though not simply confined to agricultural production (Kumar et al., 2004; Douglas et al., 2009) and drought frequency (Subbiah, 2002). Given the rugged orography of the surrounding region and difficulties in modelling intense precipitation, resolving the future roles of natural variability and the externally forced signal for the monsoon is a fundamentally difficult – but important – problem.

Interannual changes in the monsoon have been linked to internal (natural) variability inherent to the climate system. For instance, numerous studies have found a potential connection between variability in the El Niño-Southern Oscillation (ENSO) and the monsoon (Sikka 1980; Shukla and Paolino 1983; Annamalai and Liu 2005). Such links could be used to improve predictability of Indian rainfall. While internal variability likely plays a non-negligible role in modulating the South Asian summer monsoon – and is expected to continue to do so in the future, even in high emissions scenarios (Annamalai et al. 2007) – changes in the monsoon's mean state associated with external forcings are also of fundamental importance. Specifically, determining the anthropogenic impacts on monsoonal changes associated with emissions of greenhouse gases (GHGs) and aerosols can provide critical insights that can help better inform policymaking decisions regarding emission regulations.




The steady rise in GHGs over the 20[th] century has increased the atmosphere's average temperature
and water vapor content through the Clausius-Clapeyron relation, and might be expected as a result
to contribute to increased precipitation over India. CMIP6 models run with just an increase in $CO_2$
forcing exhibits such an increase uniformly across India (Figure S1). However, in reality the
picture is more complex as the literature has indicated no such observed trend for India over the
last half century (Ramesh and Goswami, 2014; Saha and Ghosh, 2019). Observed monsoon
precipitation aggregated over all of continental India (Figure 1) actually indicates a slight drying
trend over the last few decades. While these trends are not statistically significant at a 95%
confidence level, the purpose of Figure 1 is to illustrate that the increase in monsoon precipitation
expected from the growing greenhouse forcing has certainly not been detected. There may be
several mechanisms invoked to explain why Indian monsoon precipitation has not increased. Land
use changes over the Indo-Gangetic Plain have been implicated as one of the causes, where
decreased evapotranspiration may have limited the amount of available precipitable water in the
region (Paul et al., 2016). It has been shown also that aerosol effects have counterbalanced the
precipitation changes attributable to the greenhouse warming (Bollasina et al., 2011; Turner and
Annamalai, 2012). Ramanathan et al. (2005) found that aerosols over India reduce surface
shortwave radiation, which limits the amount of evaporation and thereby reduces monsoon
precipitation. Additionally, it has been shown that the atmospheric brown cloud (originally so
termed in Ramanathan and Crutzen, 2003, referring to the pervasive light absorbing aerosol layer
akin to the stratocumulus cloud decks observed over the oceans) over the Northern Indian Ocean
is associated with a stable atmosphere that limits convection. Atmospheric brown clouds consist
primarily of black and organic carbon, dust and other anthropogenic aerosols. Sources of aerosols



and their precursors in South and East Asia (indicated in Figure S2), are tied particularly to energy
production and biomass combustion, which have grown steadily in response to industrialization in
the region, though recent trends in these two regions differ. Meehl et al. (2008) similarly found
that an increased aerosol load reduced precipitation over India during the monsoon season, but that
it also increased rainfall in the pre-monsoon season. Wang et al. (2009) found that absorbing
aerosols were particularly important in influencing the summer monsoon system. The issue with
many of these studies is that they focus on individual models. There is a large spread in the
precipitation response across models reflecting differing representations of cloud and aerosol
processes (e.g. Wilcox et al., 2015), factors that may bias results given the already complex nature
of modelling precipitation over India (Ramanathan et al., 2005; Bollasina et al., 2011; Turner and
Annamalai, 2012; Ramesh and Goswami, 2014; Paul et al., 2016; Saha and Ghosh, 2019).
Multimodel ensembles can improve our understanding and help constrain uncertainty on the
impacts of different aerosol constituents on the monsoon.

Here, we analyze results from two climate model intercomparisons to better understand the
summer monsoonal impacts from sulfur and black carbon aerosols, two of the dominant
constituents of India's aerosol pollution. First, we study the Precipitation Driver Response Model
Intercomparison Project (PDRMIP; Samset et al. 2016) experiments to assess the summer
monsoon response to extreme aerosol conditions. The purpose of the PDRIMP experiments is to
determine if a precipitation signal can be detected i.e. a causal link between the emissions of sulfur
and black carbon and changes in the monsoon. Previous analysis of a set of PDRMIP experiments
which increase global BC levels tenfold found a slight enhancement in P-E during the South Asian
summer monsoon, driven by a strengthened land-sea temperature gradient (Xie at al., 2020). We



focus the first section of our analysis on Asian perturbation experiments as significant emissions
changes are expected over this region in the coming decades (e.g. Samset et al., 2019). We note
that these experiments use artificially large emission perturbations to enable isolation of signal
detection from climatic variability. Second, we study a set of regional aerosol emissions
intercomparison experiments (labeled RAEI experiments for the rest of the paper for convenience)
to assess the relative contributions of Indian and Chinese anthropogenic aerosol emissions to the
monsoon. Because remote emissions may play an important role on India's monsoon (e.g. Shawki
et al. 2018), in addition to Indian emissions we choose to study emissions from China because this
country is presently the world leading emitter of BC and $SO_2$, is in close proximity to India and its
emissions of both pollutants are expected to decline rapidly over the coming decade. Emissions in
more remote regions are less likely to change in a major way. A robust analysis of these
intercomparisons should refine our understanding of the anthropogenic influence on the South
Asian summer monsoon and reduce uncertainty on future changes given that India's anthropogenic
emissions are expected to increase at least in the near term, while China's will likely decrease (Rao
et al. 2016). We decompose precipitation changes into dynamic (i.e. circulation changes) and
thermodynamic (i.e. specific humidity changes) components to assess how aerosols interact with
the monsoon. The rest of the paper is structured as follows: section 2 discusses the simulations
used in the analysis, section 3 presents and analyzes potential monsoonal impacts associated with
sulfur and black carbon emissions and section 4 summarizes the study and highlights needs for
future work.

**2. Data and Methods**
*2.1 PDRMIP intercomparison*



We first study the Precipitation Driver Response Model Intercomparison Project (PDRMIP)
experiments. PDRMIP is an idealized set of modelling experiments, used to better understand
drivers of regional precipitation change. We focus specifically on two experiments that involve
perturbations to Asian concentrations or emissions (see Table 1), where Asia is defined by the
regional box of 60-140˚E and 10-50˚N. The first is BC10xASIA, representing a tenfold increase
in present-day BC concentrations or emissions in Asia at all vertical levels, and the second is
SULF10xASIA, which explores a similar tenfold increase in present-day sulfate concentrations or
emissions. The BC10xASIA and SULF10xASIA scenarios are compared with control simulations
(henceforth called CTRL$_{PDRMIP}$) where aerosol concentrations or emissions are maintained at near
current values (either year 2000 or 2005 for each model). We study the six models involved in the
PDRMIP experiments that conduct the Asian perturbation experiments (Table 1). These
experiments will be used to better constrain uncertainty on the direction of precipitation and
circulation changes under anthropogenic aerosol emissions changes. Since these are extreme
perturbations to aerosol concentrations, we use these scenarios not as representative of a future
emissions trajectory, but rather as a way to check if different models with different process
representations indicate a consistent response. Due to inter-model differences in spatial resolution,
all data are rescaled to the lowest model resolution ($3.75˚ \times 2.0˚$) when comparing model output.
Variations in aerosol schemes and direct and indirect aerosol effects across the six models will
affect the spread in predicted precipitation changes associated with the increased aerosol
concentrations (Table 1). The different schemes and their effects on precipitation will be discussed
further in the section 3.



**Table 1.** Details of the models analyzed in this work. For the models participating in the
PDRMIP Asian aerosol perturbation simulations, each simulation lasts 100 years. Cloud scheme
refers to the microphysical cloud scheme that describes cloud formation, where a one-moment
scheme considers only changes in mass and a two-moment scheme considers changes in mass
and number concentration. The first indirect effect refers to the aerosol effect on cloud albedo
and the second indirect effect refers to the aerosol effect on cloud lifetime.

| Model | Spatial resolution | Cloud scheme | Indirect effects | Model reference | Aerosol microphysics | MIP |
|---|---|---|---|---|---|---|
| CESM1-CAM5[†] | 1.25° × 0.9375° | Two moment | First, second | Neale et al. (2012) | Full aerosol | PDRMIP, RAEI |
| GISS-E2-R | 2.5° × 2.0° | One moment | None[*] | Schmidt et al. (2014) | No aerosol | PDRMIP, RAEI |
| HadGEM3 | 1.875° × 1.25° | One moment | First | Hewitt et al. (2011) | No BC; aerosol-cloud interaction included | PDRMIP |
| UKESM1-0-LL | 1.875° × 1.25° | Two moment | First, second | Sellar et al. (2019) | Full aerosol | RAEI |
| IPSL-CM | 3.75° × 1.875° | Two moment | First, second | Dufresne et al. (2013) | Aerosol microphysics for Twomey effect | PDRMIP |
| NorESM | 2.5° × 1.875° | Two moment | First, second | Bentsen et al. (2013) | Full aerosol | PDRMIP |
| MIROC-SPRINTARS[†] | 1.41° × 1.41° | One moment | First, second | Watanabe et al. (2011) | Full aerosol | PDRMIP |

*Indirect effects in the PDRMIP simulations were turned off since these simulations had prescribed aerosol fields
and so changes in the hydrologic cycle could not change the aerosols. The first effect was included in the GISS RAEI
simulations, however, as those are emissions-driven and hence physically consistent.
†Indicate models that change emissions in the PDRMIP experiments. Rows that do not include this mark indicate
models that change concentrations in the PDRMIP experiments.
*2.2 RAEI experiments*
The purpose of the RAEI experiments is to assess the relative contributions of aerosol emissions
from China and India on monsoon precipitation over India. Three GCMs with coupled chemistry-
climate components are used to study the effects of regional perturbations in aerosol emissions on
the Indian monsoon: GISS-E2-R (Schmidt et al., 2014), CESM1-CAM5 (Neale et al., 2012) and
UKESM1-0-LL (Sellar et al., 2019). Past research has used some of these models to explore the



173 effects of regional aerosol reductions on global precipitation, including emissions changes in the

174 US, Europe, China and India. Some of the experiments from RAEI have been used to study the

175 global effects of US $SO_2$ emissions on global precipitation (Conley et al., 2018) as well as local

176 and remote precipitation responses to regional reductions in aerosols (Westervelt et al., 2018).

177 Here, we study the South Asian summer monsoon response to reductions in anthropogenic aerosol

178 emissions in China and India, focusing specifically on a set of three experiments: (1) no $SO_2$

179 emissions in India (IND NO $SO_2$), (2) 80% $SO_2$ emissions reduction in China (CHN 20% $SO_2$)

180 and (3) no $SO_2$ emissions in India and China (IND+CHN NO $SO_2$). We have run additional BC

181 experiments that are included only in the SI because we find that changes in BC do not have a

182 clear impact on precipitation in the summer monsoon. The three $SO_2$ experiments will be compared

183 to control simulations (CTRL) with emissions set near present-day values (year 2000 or 2005

184 depending on the model) to determine the relative importance on summer monsoon precipitation

185 of regional aerosol emissions from India and China. The UKESM experiments were run over a

186 shorter period (40 years), relative to the other models (~200 years). We found from resampling

187 that 40 years is sufficient to observe the general precipitation statistics over India. For

188 climatological variables studied in our PDRMIP and RAEI analysis, we take mean values over the

189 full simulation period, excluding the first 10 years to allow for spin-up.

190

191 *2.3 Precipitation decomposition*

192 In addition to calculating overall precipitation changes due to sulfur and BC emissions, we seek

193 also to determine the dynamic and thermodynamic components of the changes attributable to these

194 forcing agents. The dynamic component is representative of precipitation changes caused by a

195 change in atmospheric circulation, and the thermodynamic component is representative of





variations in precipitation due to changes in moisture under constant circulation. To perform this
decomposition, we follow the methodology of Chadwick et al. 2016. The total precipitation change
$\Delta P$ can be expressed as
$$\Delta P = \Delta q M^* + q \Delta M^* + \Delta q \Delta M^*,$$
where q is the near-surface specific humidity and M* is a proxy for convective mass flux ($M^* =$
P/q). The first term on the right hand side is representative of thermodynamic changes ($\Delta P_{therm}$),
the second dynamic changes ($\Delta P_{dyn}$) and the third the nonlinear interaction of these two
components ($\Delta P_{cross}$). $\Delta P_{dyn}$ can be further decomposed into shifts in the circulation patterns
($\Delta P_{shift}$) and changes in the mean strength of the tropical circulation ($\Delta P_{strength}$) as
$$\Delta P_{shift} = q \Delta M^*_{shift},$$
$$\Delta P_{shift} = q \Delta M^*_{strength},$$
where $\Delta M^*_{strength} = -\alpha M^*_{strength}$ (where $\alpha$ = tropical mean $\Delta M^*$/tropical mean $M^*$). $\Delta M^*_{shift}$ is
computed as the residual of $\Delta M^*$ and $\Delta M^*_{strength}$. This decomposition follows the methodology in
Chadwick et al. 2016 and Monerie et al. 2019.

**3. Results**
*3.1 PDRMIP analysis: summertime Indian precipitation response to large BC and sulfur*
*perturbations*
We start with an evaluation using the PDRMIP experiments (Table 1) of summertime Indian
precipitation changes caused by large BC and sulfate concentration increases over all of Asia. The
difference in summer precipitation between the BC10xASIA and CTRL$_{PDRMIP}$ experiments
provides an estimate for the role of BC in monsoonal changes and is shown in Figures 2a-g. From
the individual models (Figures 2a-f), there is a noticeably large ensemble spread in the



precipitation response over India due to the increase in BC. In north India, for example, HadGEM3
shows a precipitation decrease of up to 70%, while SPRINTARS exhibits effectively a null
response and GISS is identified with a strong precipitation increase of ~50%. PDRMIP simulations
that globally increase BC tenfold also do not show a consistent multimodel response over India
(Samset et al. 2016; Liu et al 2018). While HadGEM3 and GISS generally underestimate
precipitation over India (Figure S3), it does not appear that these biases are manifest in consistent
precipitation changes in the BC10xASIA experiments. Additionally, while two of the six models
studied increase BC emissions rather than BC concentrations, this does not appear to alter the BC
vertical profile except in the stratosphere (see Figure S4). It is likely that different aerosol schemes
across models (Table 1) may be implicated as the dominant source of the large ensemble spread,
although both the boundary layer scheme and modelling impacts of absorbing aerosols on cloud
formation (Koch and Del Genio, 2010) could play important roles. Specifically, cloud formation
is affected significantly by the BC vertical profile; BC within the cloud layer can burn off moisture
and reduce cloud cover, BC below the cloud layer can enhance convection and increase cloud
cover and BC above the cloud layer can either increase or decrease cloud cover according to the
cloud type. Because of the complexities of the semi-direct effects of absorbing aerosols that are
currently not heavily constrained by observations, the role of BC generally has a diverse response
across climate models (Koch et al., 2009; Stjern et al. 2017). Large variance in the cloud fraction
vertical profile are apparent also in the PDRMIP BC10xASIA simulations (Figure 3). This large
uncertainty does not consistently favor an increase or decrease in cloud fraction across vertical
layers except in NorESM and CESM where a slight increase (on the order of a couple of percent)
can be detected across all layers. Variations in the BC vertical profile as well as its lifetime can
result in significant changes in cloud cover and precipitation even within an individual model by





changing atmospheric stability and humidity (Samset and Myhre 2015). These effects are manifest
in the diverse shortwave responses (Figure S5), which indicate a large spread between models in
magnitude and sign over parts of India. Additionally, changes in the TOA net radiative forcing
between BC10xASIA and PDRMIP$_{CTRL}$ are generally consistent in magnitude and direction across
models over India (Figures S7a-f). By contrast, the change in Cloud Radiative Effect (CRE;
Figures S7g-l) is not consistent in sign across models, suggesting that the models agree on the
direct aerosol effects but differ on the aerosol-cloud interactions. While there are more causative
factors on precipitation than cloud fraction, the important point is that because of the large cloud
uncertainty that varies in both magnitude and sign, it is difficult to attribute future changes in
Indian precipitation to changes in BC concentration. This is reflected in the precipitation change
which fails to demonstrate a clear spatial coherence in the multimodel mean (Figure 2g).

The role of sulfate for Indian precipitation is much clearer. The percent change in precipitation
between the SULF10xASIA and CTRL PDRMIP experiments is shown in Figures 2h-n. The sign
of the precipitation change is generally consistent across models, with a large decrease in
precipitation (~50%) over all of India in response to a tenfold increase in sulfate. There is also
large uncertainty in the cloud fraction profile response to sulfate emissions (Figure 3), similar to
the BC PDRMIP experiments. However, five of the six models on average favor a decrease in
cloud fraction with increased SO$_2$ emissions, consistent with the precipitation response. So, while
there is a comparable measure of intermodel spread for the BC10xASIA and SULF10xASIA cloud
responses, the mean change is more consistent in the SULF10xASIA experiments. The results
from the PDRMIP experiments, with their higher sulfate concentrations, constrain uncertainty on





the sign of precipitation changes, and can be used as a frame of reference for the country-specific
aerosol experiments described in section 3.2 and beyond.

*3.2 RAEI analysis: Indian aerosol burden response to Chinese and Indian aerosol emissions*
*changes*
We now consider the RAEI emissions scenarios for China and India. Percent changes in sulfate
burden between the sulfate regulation scenarios and control runs are shown in Figures S7a-i. Indian
sulfate emissions play an important role on local sulfate concentrations, contributing up to 60% of
the country's aerosol burden, while China's emissions can contribute up to 60% over the
Himalayas. The change in Indian aerosol burden for sulfate is notably consistent in terms of both
the magnitude of the change as well as the spatial pattern across the three models studied. Since
the temperature gradient between the Arabian Sea and Bay of Bengal and the Himalayas has been
invoked as a modulator of the South Asian Monsoon (e.g. Priya et al., 2017), both Indian and
Chinese emissions could influence monsoon precipitation over India by modifying the optical
properties of the atmosphere not only over the country but also over surrounding regions.

*3.3 RAEI analysis: summer monsoon precipitation response to regional $SO_2$ emissions changes*
The precipitation response associated with $SO_2$ emissions is significant over parts of India (Figures
4a-i), in agreement with the PDRMIP results. Almost all models and scenarios show an increase
in summer precipitation in India when $SO_2$ emissions in China and/or India are reduced. The
strongest response requires regulations from both China and India, with an increase of nearly 20%
in precipitation in some regions of India when $SO_2$ emissions are reduced across the three models
studied here. From these results, changes in India's precipitation depend not only on local $SO_2$





emissions, but also on regional sources. These emissions can have a measurable impact on India's
water availability by altering the underlying statistics in favor of greater precipitation events (e.g.
Sillman et al. 2019). That being said, the spatial patterns associated with these precipitation
changes vary to a large degree between models. For instance, precipitation changes in GISS exhibit
greater consistency across scenarios than they do with the CESM or UKESM. Additionally,
UKESM tends to estimate larger precipitation changes than the other RAEI models, consistent
with the HadGEM3 results indicated in Figure 2 which uses the same physical model. There is,
however, general consistency in the increase in precipitation when $SO_2$ emissions are reduced in
both China and India. The precipitation responses to lower BC regional emissions are indicated in
Figure S8. BC emissions play a much lesser role in GISS and CESM relative to $SO_2$ emissions,
and cause an inconsistent response in UKESM across the three regional emissions experiments.
For all reduced BC scenarios, the changes in India's precipitation are generally small (~5% locally)
and not statistically significant at a 90% confidence level. The strongest precipitation response
occurs when both Chinese and Indian BC emissions are eliminated, but there is a spread in the
direction of change across models. This spread in precipitation change is consistent with that of
the PDRMIP results in that the intermodel spread in precipitation change due to BC emissions
changes tends to be larger than the magnitude of the precipitation response from any individual
model. This may highlight large process uncertainty generally. Bond et al. (2013), for example,
note that the impact of BC on the cloud radiative forcing in models is highly sensitive to the
nucleation regime in the background atmosphere.

*3.4 RAEI analysis: physical understanding of the $SO_2$-precipitation response*





Physical explanations for the precipitation changes induced by $SO_2$ emissions changes are
explored here. Circulation changes are typically connected to sulfate increases in India; a
weakened land-sea temperature gradient associated with $SO_2$ emissions would inhibit monsoonal
advection of moisture from the Arabian Sea onto the Indian subcontinent. Warming over the
Himalayas can be seen in most of the simulations (Figure S9), as well as changes in 850 hPa winds,
where there is a clear strengthening of the coastal winds when $SO_2$ emissions are reduced (Figure
4). The fact that the land-sea temperature gradient and 850 hPa winds change suggests that
precipitation changes due to $SO_2$ emissions may be dynamically rather than thermodynamically
driven, which motivates the precipitation decomposition analysis discussed later. As shown in
Figure 4, strengthening of the monsoonal winds is largely consistent across models and scenarios,
though there are slight differences in the location of the strongest zonal wind increases; in GISS
and UKESM, the greatest increase is over India itself for most scenarios, while it is further south
in CESM. This suggests that a high sulfate burden reduces the strength of the monsoon winds,
consistent with prior studies that connect these changes to the dimming of the downward solar flux
(Kim et al. 2007). The relative contributions of thermodynamic (i.e. specific humidity) changes to
dynamic (i.e. circulation) changes are indicated in Figure 5. The thermodynamic precipitation
response to sulfur emissions reductions is positive for the three emissions experiments, consistent
with the Clausius-Clapeyron relation as less $SO_2$ increases surface temperatures and consequently
specific humidity. The interaction of dynamic and thermodynamic components (panel c, $\Delta P_{cross}$)
plays a minimal role. The magnitude of the thermodynamic response is on the order of 50% that
of the dynamic component – i.e. the dynamic component dominates. Panels (d) and (e) of Figure
5 indicate that this effect is driven primarily by shifts in the convective regions, with changes in
the tropical mean circulation having a minimal or slightly negative effect. It is of note that the



magnitude of each component is consistent across the three models studied here, suggesting
consistency in the mechanistic reasons for increased monsoon precipitation over India when sulfur
emissions are reduced. Changing circulation patterns are suggested as a consequence of changes
in $CO_2$ as well, and potential nonlinear effects of sulfur and greenhouse emissions on monsoon
precipitation highlight an important point that demands further study.

**4. Conclusions**
The main purpose of this study was to better understand, through the use of several GCM
experiments, the sensitivity of the South Asian summer monsoon to regional anthropogenic aerosol
emission changes. Given that this is a modelling study, there are a number of caveats that must be
acknowledged. There are often questions of how well GCMs can simulate the Indian monsoon
since their spatial resolution may be too coarse to resolve the complex orography of India and the
surrounding regions (Prell and Kutzbach, 1992). Additionally, representation of cloud
microphysical processes is a known limitation of GCMs (e.g. Wilcox et al., 2015). We find a large
intermodel spread in cloud profile and precipitation changes in the various BC emissions scenarios
studied here. This suggests that discrepancies in representation of cloud processes within GCMs
constrain uncertainty in the precipitation response from BC perturbations, which cannot be
accounted for simply by differences in the BC vertical profiles (Figure S4). In contrast, the
precipitation responses for $SO_2$ emission changes as well as the dynamic mechanism for these
responses are largely consistent across models, suggesting that there is relative certainty in the
models ability to simulate precipitation changes due to $SO_2$ emissions. So, while it may be difficult
to extrapolate on the basis of these simulations from modelled to real-world monsoon precipitation





changes induced by anthropogenic aerosols, consistency in the $SO_2$ response across models lends
confidence in a potential observed response for future emissions changes.

On investigating the response of the monsoon to a tenfold increase of Asian BC and sulfate
concentrations, we found that the role of BC on Indian precipitation is uncertain but that increased
sulfate concentrations over India reduce precipitation across five of the six models studied. Large
uncertainty in the precipitation response to changing Asian BC is notably consistent with previous
PDRMIP analysis studying monsoon changes to a tenfold increase in global BC levels (Xie et al.
2020). Consistency between the global and regional PDMRIP simulations in this context suggests
further that a BC signal is difficult to detect for the South Asian summer monsoon.

When assessing the relative contributions of Chinese and Indian anthropogenic $SO_2$ emissions to
aerosol loading over South Asia (the RAEI emissions experiments), and the consequent
precipitation responses, we find that there is only a statistically significant difference in monsoon
precipitation when there is regulation of both China and India's $SO_2$ emissions, which leads to on
the order of a 20% precipitation increase locally. Consistency in the precipitation responses
between the increased sulfate scenario (PDRMIP SULF10xASIA) and the decreased sulfate
scenario (RAEI) suggests that the aerosol-precipitation link may be a reversible process, and is
attributable in large part to dynamical changes specifically shifts in convective patterns over the
region. Additionally, these results are significant because Chinese emissions of $SO_2$ have declined
over the past decade, while Indian emissions have grown steadily. There is also anticipated growth
in $CO_2$ emissions and concentrations over the coming decades and this is expected to result in an
increase in the atmospheric water vapor content. These concurrent events will have important



implications for policy going forward, as water deficits present a major issue for India that may be
exacerbated given the country's exponential population growth. Regions that exhibit large
variability in summertime precipitation such as Chennai and Delhi (as indicated in Figure S10)
may be particularly sensitive to future monsoon changes because interannual shifts between wet
and dry years at present impose important strains on the available water resource. Moreover, the
benefits of policies to control $SO_2$ emissions will have significant impacts not only on mitigating
water deficits but also in terms of alleviation of air pollution, estimated to be responsible for
hundreds of thousands of premature deaths per year in India (Health Effects Institute, 2019).

While China's pollution is expected to decline in most socio-economic projections, India's is
expected to grow except under strong emissions controls (Samset et al., 2019). Regardless of the
realism of these scenarios, the results should be seen as further impetus for regional policies to
reduce $SO_2$ emissions given that we have found combined emissions reductions from China and
India can increase monsoon precipitation over the country by 5% on average and by up to 20%
locally. This effect, in combination with consequent impacts of continued growth in GHGs (Figure
S1), could result in an overabundance. This calls therefore for careful consideration of implications
for both precipitation and health over multiple timescales.

**Code and data availability**
All code and model data to make the figures used in this paper will be made publicly available
through Zenodo following acceptance of the paper. The ESRL database makes gridded
precipitation data publicly available for both the. University of Delaware data



(https://www.esrl.noaa.gov/psd/data/gridded/data.UDel_AirT_Precip.html) and for the GPCC
data (https://www.esrl.noaa.gov/psd/data/gridded/data.gpcc.html).

**Author contribution**

ATA, NLA, JFL, DS, GF ran the RAEI experiments for their respective GCMs. PS prepared the
manuscript with contributions from all co-authors.

**Competing interests**

The authors declare that they have no conflict of interest.

**Acknowledgments**

This study was supported by the Harvard Global Institute. ATA and NLA thank NERC through
NCAS for funding for the ACSIS project and NE/P016383/1. The UKESM work used Monsoon2,
a collaborative High Performance Computing facility funded by the Met Office and the Natural
Environment Research Council. This work used JASMIN, the UK collaborative data analysis
facility. The NCAR-CESM work is supported by the National Science Foundation and the Office
of Science (BER) of the U.S. Department of Energy. NCAR is sponsored by the National Science
Foundation. Climate modeling at GISS is supported by the NASA Modeling, Analysis and
Prediction program. GISS simulations used resources provided by the NASA High-End
Computing (HEC) Program through the NASA Center for Climate Simulation (NCCS) at Goddard
Space Flight Center.



Annamalai, H. and Liu, P., 2005: Response of the Asian Summer Monsoon to changes in El Niño

properties. *Quart. J. Roy. Meteor. Soc.*, **131,** 805-831.

Annamalai, H., Hamilton, K. and Sperber, K.R., 2007: South Asian summer monsoon and its

relationship with ENSO in the IPCC AR4 simulations. *J. Clim.*, **20**, 1071-1083.

Bentsen, M., et al., 2013: The Norwegian Earth System Model, NorESM1-M – Part 1: Description

and basic evaluation of the physical climate. *Geosci. Model Dev.*, **6**, 687-720.

Bollasina, M.A., Ming, Y. and Ramaswamy, V., 2011: Anthropogenic aerosols and the weakening

of the South Asian Summer Monsoon. *Science*, 6055(**334**), 502-505.

Bond, T.C., et al., 2013: Bounding the role of black carbon in the climate system: A scientific

assessment. *J. Geophys. Res.-Atmos.*, **118**, 5380-5552.

Chadwick, R., Good, P., and Willett, K.M., 2016: A simple moisture advection model of specific

humidity change over land in response to SST warming. *J. Clim.*, **29**, 7613–7632.

Conley, A.J., et al., 2018: Multimodel surface temperature responses to removal of US sulfur

dioxide emissions. *J. Geophys. Res.-Atmos.*, **123**, 2773-2796.

Douglas, E.M., Beltrán-Przekurat, A., Niyogi, D., Pielke, R.A. and Vörösmarty, C.J., 2009: The

impact of agricultural intensification and irrigation on land–atmosphere interactions and

Indian monsoon precipitation – a mesoscale modeling perspective. *Glob. Planet.*

*Change*, **67**, 117-128.

Dufresne, J.-L., et al., 2013: Climate change projections using the IPSL-CM5 Earth System Model:

from CMIP3 to CMIP5. *Clim. Dyn.*, 10(**40**), 2123-2165.

Eyring, V., Bony, S., Meehl, G.A., Senior, C.A., Stevens, B., Stouffer, R.J. and Taylor, K.E., 2016:

Overview of the Coupled Model Intercomparison Project Phase 6 (CMIP6) experimental

design and organization. *Geosci. Model Dev.*, **9**, 1937-1958.





Health Effects Institute, 2019: State of Global Air 2019. Data source: Global Burden of Disease

Study 2017. *IHME*, 2018.

Hewitt, H.T., et al., 2011: Design and implementation of the infrastructure of HadGEM3: the next-

generation Met Office climate modelling system. *Geosci. Model Dev.*, **4**, 223-253.

Kim, M.-K., Lau, W.K.M., Kim, K.-M. and Lee, W.-S., 2007: A GCM study of effects of radiative

forcing of sulfate aerosol on large scale circulation and rainfall in East Asia during boreal

spring. *Geophys. Res. Lett.*, **34**, L24701.

Koch, D. and Del Genio, A.D., 2010: Black carbon semi-direct effects on cloud cover: review and

synthesis. *Atmos. Chem. Phys.*, **10**, 7685-7696.

Koch, D., et al., 2009: Evaluation of black carbon estimations in global aerosol models. *Atmos.*

*Chem. Phys.*, **9**, 9001-9026.

Kumar, K.K., Kumar, R.K., Ashrit, R.G., Deshpande, N.R. and Hansen J.W., 2004: Climate

impacts on Indian agriculture. *International J. of Clim.*, **24**, 1375-1393.

Liu, L., et al., 2018: A PDRMIP multimodel study on the impacts of regional aerosol forcings on

global and regional precipitation. *J. of Clim.*, **31**, 4429-4447.

Meehl, G.A., Arblaster, J.M. and Collins, W.D., 2008: Effects of black carbon aerosols on the

Indian Monsoon. *J. Clim.*, **21**, 2869-2882.

Monerie, P.-A., Robson, J., Dong, B., Hodson, D. L. R., & Klingaman, N. P. (2019). Effect of the

Atlantic multidecadal variability on the global monsoon. *Geophys. Res. Lett.*, **46**, 1765–

1775.

Neale, R.B., et al., 2012: Description of the NCAR Community Atmosphere Model (CAM 5.0).

NCAR Tech. Note TN-486, 274 pp.



Paul, S. et al., 2016: Weakening of Indian summer monsoon rainfall due to changes in land use

land cover. *Sci. Rep.*, **6**, 32177.

Prell, W.L., and Kutzbach, J.E., 1992: Sensitivity of the Indian monsoon to forcing parameters and

implications for its evolution, *Nat.*, **360**, 647–652.

Priya, P., Krishnan, R., Mujumdar, M., and Houze Jr., R.A., 2017: Changing monsoon and

midlatitude circulation interactions over the Western Himalayas and possible links to

occurrences of extreme precipitation. *Clim. Dyn.*, **49**, 2351-2364.

Ramanathan, V. and Crutzen, P., 2003: New directions: Atmospheric brown "clouds". *Atmos.*

*Env.*, **37**, 4033-4035.

Ramanathan, V., et al., 2005: Atmospheric brown clouds: Impacts on South Asian climate and

hydrological cycle. *PNAS*, 102(**15**), 5326-5333.

Ramesh, K.V. and Goswami, P., 2014: Assessing reliability of climate projections: the case of

Indian monsoon. *Sci. Rep.*, **4**, 161-174.

Rao, S., et al., 2017: Future air pollution in the Shared Socio-economic Pathways, *Global Environ.*

*Chang.*, **42**, 346-358.

Saha, A. and Ghosh, S., 2019: Can the weakening of Indian Monsoon be attributed to

anthropogenic aerosols? *Environ. Res. Commun.*, **1**, 061006.

Samset, B.H. and Myhre, G., 2015: Climate response to externally mixed black carbon as a

function of altitude. *J. Geophys. Res.*, **120**, 2913-2927.

Samset, B.H, et al., 2016: Fast and slow precipitation responses to individual climate forcers: A

PDRMIP multimodel study. *Geophys. Res. Lett.*, **43**, 2782-2691.

Samset, B.H., Lund, M.T., Bollasina, M., Myhre, G. and Wilcox, L., 2019: Emerging Asian

aerosol patterns. *Nat. Geosci.*, **12**, 582-584.



Schmidt, G.A., et al., 2014: Configuration and assessment of the GISS ModelE2 contributions to

the CMIP5 archive. *J. Adv. Model. Earth Syst.*, 1(**6**), 141-184.

Schneider, U., Becker, A., Finger, P., Meyer-Christoffer, A. and Ziese, M., 2018: GPCC Full Data

Monthly Product Version 2018 at 0.5°: Monthly Land-Surface Precipitation from Rain-

Gauges      built      on      GTS-based      and      Historical      Data.      DOI:

10.5676/DWD_GPCC/FD_M_V2018_050

Sellar, A.A., et al., 2019: UKESM1: Description and evaluation of the UK Earth System Model.

*J. Adv. Model. Earth Syst.*, **11**.

Shawki, D., Voulgarakis, A., Chakraborty, A., Kasoar, M., and Srinivasan, J., 2018: The South

Asian Monsoon response to remote aerosols: global and regional mechanisms. *J. Geophys.*

*Res.*, **123**, 11,585-11,601.

Shukla, J. and Paolino, D.A., 1983: The Southern Oscillation and long-range forecasting of the

summer monsoon rainfall over India. *Mon. Wea. Rev.*, **111**, 1830-1837.

Sillmann, J., et al., 2019: Extreme wet and dry conditions affected differently by greenhouse gases

and aerosols. *npj Clim. Atmos. Sci.*, **2**, 24.

Sikka, D.R., 1980: Some aspects of the large-scale fluctuations of summer monsoon rainfall over

India in relation to fluctuations in the planetary and regional scale circulation

parameters. *Proc. Indian Natl. Acad. Sci.*, **89,** 179-195.

Stjern, C.W., et al. 2017: Rapid adjustments cause weak surface temperature response to increased

black carbon concentrations. *J. Geophys. Res.*, **122**, 11,462-11,481.

Subbiah, A. *Initial Report on the Indian Monsoon Drought of 2002* (Asian Disaster Preparedness

Center, 2002).



Turner, A.G. and Annamalai, H., 2012: Climate change and the South Asian Summer Monsoon.

*Nat. Clim. Change*, **2**, 1-9.

Wang, C., Kim, D., Ekman, A.M.L., Barth, M.C., Rasch, P.J., 2009: Impact of anthropogenic

aerosols on Indian summer monsoon. *Geophys. Res. Lett.*, **36**, L21704.

Watanabe, S., et al., 2011: MIROC-ESM 2010: Model description and basic results of CMIP5-

20c3m experiments. *Geosci. Model Dev.*, **4**, 845-872.

Westervelt, D.M., et al., 2018: Connecting regional aerosol emissions reductions to local and

remote precipitation responses. *Atmos. Chem. Phys.*, **18**, 12461-12475.

Wilcox, L.J., Highwood, E.J., Booth, B.B.B. and Carslaw, K.S., 2015: Quantifying sources of

inter-model diversity in the cloud albedo effect. *Geophys. Res. Lett.*, 42(**5**), 1568-1575.

Willmott, C.J. and Matsuura, K., 2001: Terrestrial Air Temperature and Precipitation: Monthly

and        Annual        Time        Series        (1950        -        1999),

http://climate.geog.udel.edu/~climate/html_pages/README.ghcn_ts2.html.

Xie, X., et al., 2020: Distinct responses of Asian summer monsoon to black carbon aerosols and

greenhouse gases. *Atmos. Chem. Phys. Discuss.*, in review.




**Figures**

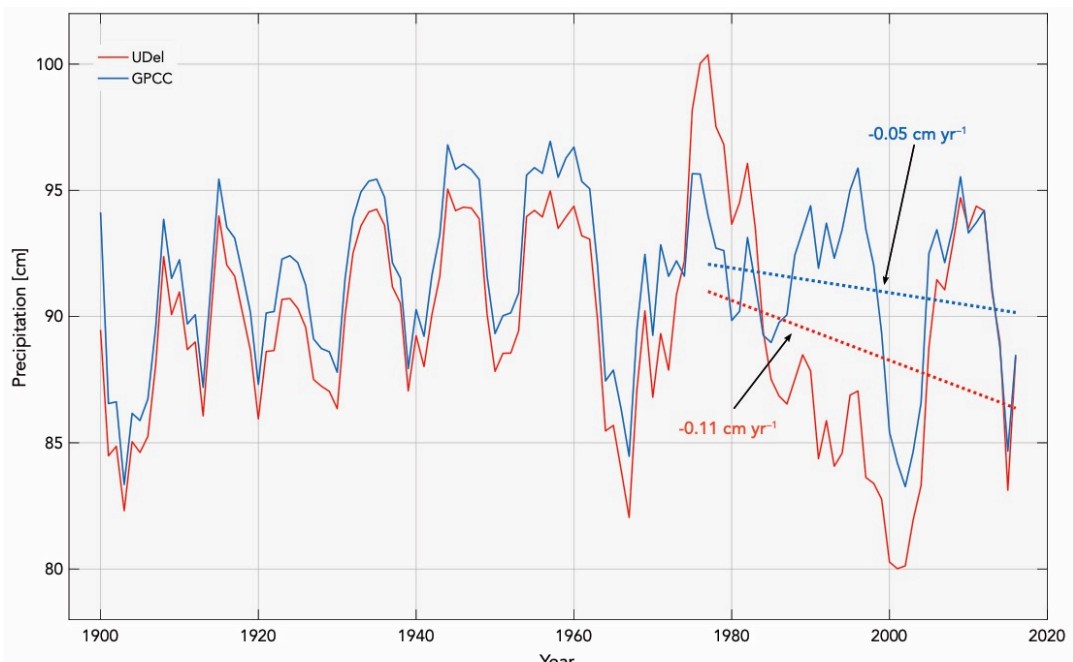

**Figure 1.** Average cumulative summer (JJAS) precipitation [cm] over land in all of India from
1900 to 2016 for two observational datasets: (red) University of Delaware (UDel; Willmot and
Matsuura, 2001) (blue) the Global Precipitation Climatology Center (GPCC; Schneider et al.
2018). Data are smoothed using a moving mean with a window size of five years. Linear trend
lines are indicated for the last 40 years for each dataset as dashed lines, and the slopes [cm yr$^{-1}$]
are denoted by the arrows.



**BC10xASIA – CTRL<sub>PDRMIP</sub>**

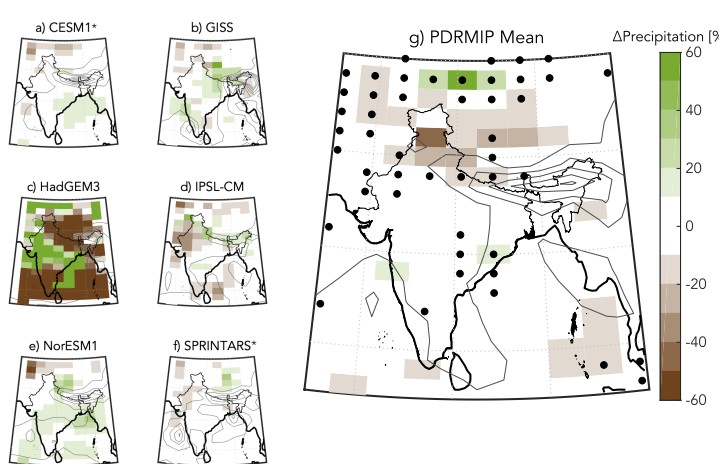

**SULF10xASIA – CTRL<sub>PDRMIP</sub>**

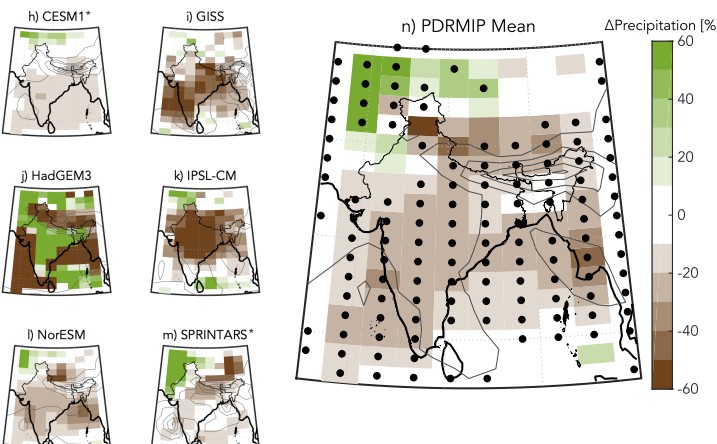


**Figure 2.** Percent change in summertime (JJAS) precipitation between (a-f) the BC10xASIA and
the CTRL$_{PDRMIP}$ runs; (g) the multimodel mean of the change. Similarly, (h-m) represent the
precipitation change in JJAS precipitation between the SULF10xASIA scenarios and the
CTRL$_{PDRMIP}$ runs, and (n) represents the multimodel mean of the change. Stippled grid cells in
(g) and (n) denote regions where at least five of the six models agree on the sign of the change.
Grey contours indicate mean JJAS precipitation from the control experiment for each model at 5
mm day$^{-1}$ intervals.

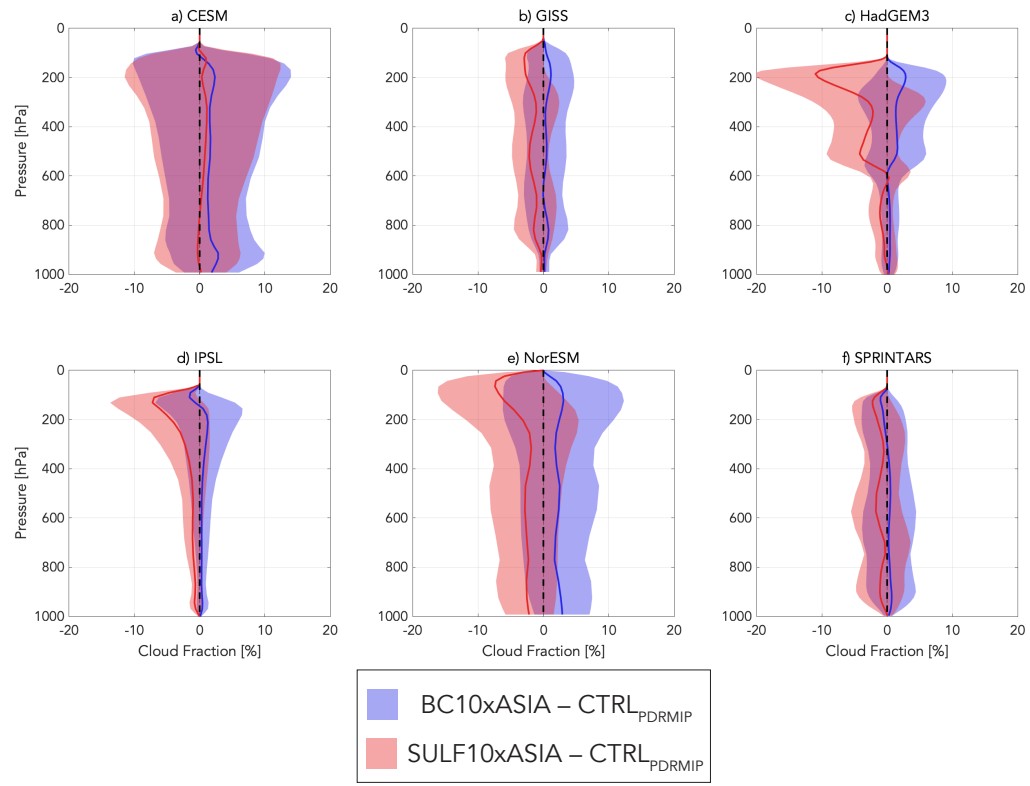


**Figure 3.** JJAS difference in cloud fraction between (blue) the BC10xASIA and the CTRL$_{PDRMIP}$
runs and (red) the SULF10xASIA scenarios and the CTRL$_{PDRMIP}$ runs. The bold lines represent
the mean difference and the shadings represent 25$^{th}$ and 75$^{th}$ percentiles.


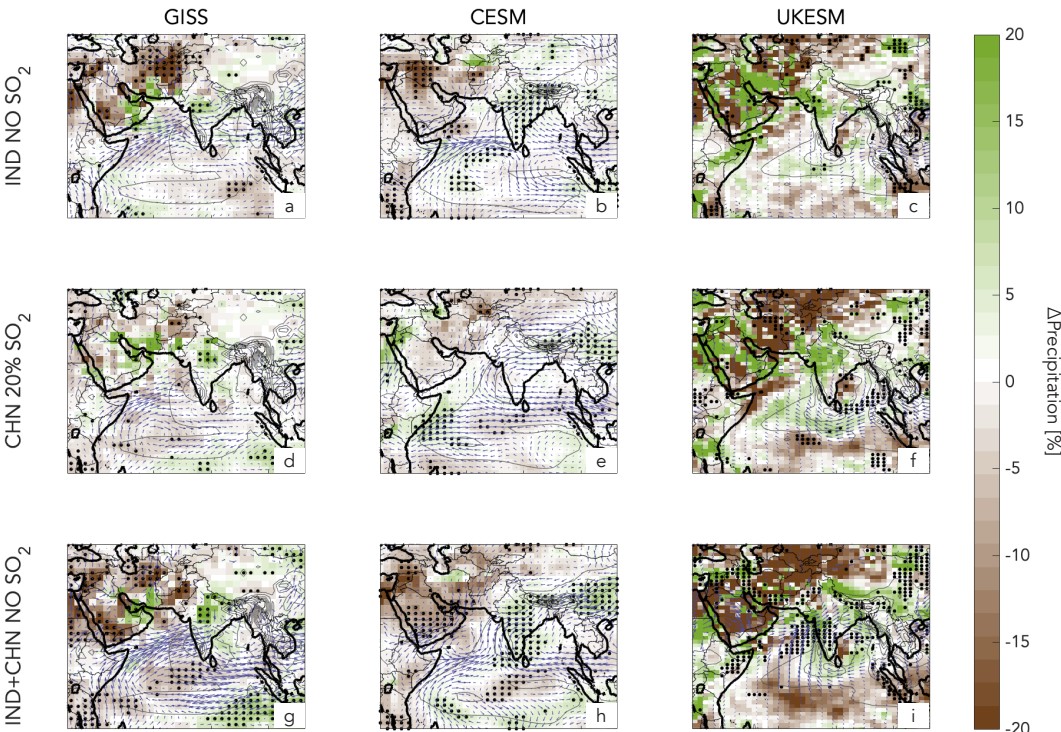

**Figure 4.** JJAS precipitation percentage difference between the SO₂ regional emissions scenarios
and the CTRL runs. JJAS 850 hPa wind changes are overlaid for each simulation. The columns
represent the different models and rows represent the different emissions scenarios. Stippled
regions denote areas where the difference is significant at a 90% confidence level for a two-
sample t-test. Grey contours indicate mean JJAS precipitation from the control experiment for
each model at 5 mm day$^{-1}$ intervals.




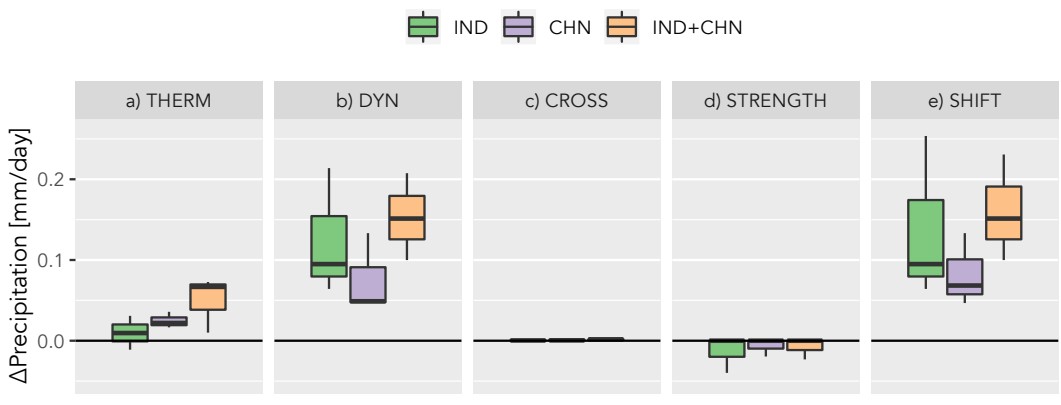


**Figure 5.** Boxplots indicating the decomposition of area averaged JJAS precipitation anomalies
[mm day$^{-1}$] into a) $\Delta P_{therm}$, b) $\Delta P_{dyn}$, c) $\Delta P_{cross}$, d) $\Delta P_{strength}$ and e) $\Delta P_{shift}$ components over India.
Different colors represent the three RAEI scenarios relative to the respective CTRL run with
green representing the IND NO $SO_2$ experiment, purple the CHN 20% $SO_2$ experiment and
orange the IND+CHN NO $SO_2$ experiment. The range for each boxplot corresponds to
intermodel variability from the three different models studied in the RAEI experiments.