# Peer review of "Sensitivity of modeled Indian Monsoon to Chinese and Indian aerosol"

_Atmospheric Chemistry and Physics, 2020_

## Referee Comment (RC1) · Matthew Kasoar (Referee) · 15 Nov 2020

This paper presents a detailed analysis of the Indian monsoon precipitation response to both idealised and more realistic regional aerosol perturbations, conducted with several different current-generation composition-climate models. The paper is notable for bringing together simulations from several different models in its analysis. The multi-model aspect of this study allows identification of robust features of the response which represents a significant update on previous literature which have studies this problem typically with single models at a time.

As a result, I would recommend that this paper be accepted for publication in ACP, subject to satisfactorily addressing a handful of minor comments/corrections I have

below, which I hope should all be straightforward.

Minor comments:

- Figure S1: I'm a little curious about the stippling in the multi-model mean Fig S1(l) - the caption says it indicates where >70% of the models agree on the sign. However, all except one of the individual models seem to show +ve responses in the south-west corner, and yet this is the only bit of panel (l) which isn't stippled... Conversely, several of the models seem to disagree about the response in the north-east corner, but panel (l) shows stippling here. Could the authors just double-check that the stippling has been applied correctly, to set my mind as ease.

- Also on Figure S1, and the associated discussion in L65-68: I'm surprised by the magnitude of the response by up to 30-40% in some models. The caption suggests that this is after 30 years of 1% per year CO2 increase on pre-industrial concentrations, which would take the simulations up to ~380ppm, slightly below present-day levels, at which you would expect a little under 1K warming. Consequently, this precipitation increase would seem to be far greater than what can be explained purely from the Clausius-Clapeyron relationship. Are there additional factors contributing to this?

- Figure S2: What is the source for this emissions dataset? The caption says that it is a 2000-2015 average of the input emissions to the PDRMIP and RAEI experiments, but the methods section indicates that both these sets of simulations used timeslice 2000 or 2005 emissions.

- Table 1: The info for the indirect effects for HadGEM3 and IPSL seems to contradict the equivalent Table 1 of Liu et al., J. Clim. (2018), which describes the PDRMIP regional experiments. According to Liu et al., HadGEM3 includes both 1st and 2nd indirect effects for HadGEM3 sulfate, whilst IPSL includes 1st indirect effect only, which differ from the descriptions given here. Please double check the info.

- The present study uses 6 PDRMIP models (L142 and Table 1), but Liu et al. (2018)

describes 7 PDRMIP models that ran the regional aerosol experiments. Why was CESM1-CAM4 not included in the present study, given that this model had apparently also run the PDRMIP regional experiments?

- Methods and Results sections: On that point, the Liu et al. (2018) is not properly credited in this manuscript. The Liu et al. (2018) paper describes and presents initial analysis of the precipitation response to the PDRMIP regional experiments, including a brief discussion of the Asian monsoon response. This by no means detracts from the present study, which provides a much more in depth analysis of the Indian monsoon response in these PDRMIP experiments, however Liu et al. needs to be appropriately cited. Currently the only place I can find it referenced is in L223 where it is incorrectly referenced with regard to the global PDRMIP experiments, when in actual fact this paper analysed the regional experiments, and was the first to do so.

- L246-247: I think this should say Figure S6 not S7

- L282: "Almost all models" - there's only three models, so maybe just say "2 out of 3", otherwise it sounds more confident than it really is

- Section 3.3/3.4: The authors could consider also referencing Shawki et al., JGRA (2018) in the discussion here, which found the same response of increased Indian monsoon precip in response to reducing Chinese SO2 emissions, using HadGEM3 (precursor to UKESM1), and attributed this to the increased land-sea temperature contrast. This supports your results here, and it could be good to note the consistency with this previous study

- L298-299: "For all reduced BC scenarios, the changes in India's precipitation are generally small (∼5% locally) and not statistically significant at a 90% confidence level" - looking at Figure S8, this statement doesn't seem to be true for UKESM1

- L361-362: Again Liu et al. (2018) should also be referenced here, as it previously showed this for the regional Asian BC PDRMIP experiment as well

References: Liu, L., and Coauthors, 2018: A PDRMIP Multimodel Study on the Impacts of Regional Aerosol Forcings on Global and Regional Precipitation. J. Climate, 31, 4429–4447, https://doi.org/10.1175/JCLI-D-17-0439.1.

Shawki, D., Voulgarakis, A., Chakraborty, A., Kasoar, M., & Srinivasan, J. (2018). The South Asian monsoon response to remote aerosols: Global and regional mechanisms. Journal of Geophysical Research: Atmospheres, 123. https://doi.org/10.1029/2018JD028623

---

## Author Comment (AC1) · 27 Jan 2021

**Title: Sensitivity of modeled Indian Monsoon to Chinese and Indian aerosol emissions**

**_Response to Review #1_**

_**We would like to thank the editor and reviewers for the helpful comments and for the thorough review of our paper.**_

We thank the reviewer for the careful review of our paper. We have addressed reference issues and questions from Reviewer #1. Detailed responses to the reviewer is given below. All of the changes in the manuscript are indicated in red in the Additional Material file.

*Reviewer comments to the author:*

*This paper presents a detailed analysis of the Indian monsoon precipitation response to both idealised and more realistic regional aerosol perturbations, conducted with several different current-generation composition-climate models. The paper is notable for bringing together simulations from several different models in its analysis. The multimodel aspect of this study allows identification of robust features of the response which represents a significant update on previous literature which have studies this problem typically with single models at a time. As a result, I would recommend that this paper be accepted for publication in ACP, subject to satisfactorily addressing a handful of minor comments/corrections I have below, which I hope should all be straightforward.*

We thank the reviewer for the helpful comments throughout the review process. We have addressed the comments and corrections suggested by the reviewer here.

*1. Figure S1: I'm a little curious about the stippling in the multi-model mean Fig S1(l) - the caption says it indicates where >70% of the models agree on the sign. However, all except one of the individual models seem to show +ve responses in the south-west corner, and yet this is the only bit of panel (l) which isn't stippled... Conversely, several of the models seem to disagree about the response in the north-east corner, but panel (l) shows stippling here. Could the authors just double-check that the stippling has been applied correctly, to set my mind as ease.*

We thank the reviewer for pointing out this issue, and have fixed the stippling issue.

*2. Also on Figure S1, and the associated discussion in L65-68: I'm surprised by the magnitude of the response by up to 30-40% in some models. The caption suggests that this is after 30 years of 1% per year $CO_2$ increase on pre-industrial concentrations, which would take the simulations up to ~380ppm, slightly below present-day levels, at which you would expect a little under 1K warming. Consequently, this precipitation increase would seem to be far greater than what can be explained purely from the Clausius-Clapeyron relationship. Are there additional factors contributing to this?*

We thank the reviewer for pointing this out. There was an issue in the code for this figure and the actual change in precipitation was smaller in magnitude. We switched this figure to the 4x$CO_2$ CMIP6 runs in order to get a larger signal. The magnitude of these changes can be very large in some locations (~60% for some models), which may be due in part to Clausius-Clapeyron, but also to a shift in the strength and position of the ITCZ.

*3. Figure S2: What is the source for this emissions dataset? The caption says that it is a 2000-2015 average of the input emissions to the PDRMIP and RAEI experiments, but the methods section indicates that both these sets of simulations used timeslice 2000 or 2005 emissions.*

We thank the reviewer for making this point. The emissions data used in this figure are the black carbon and sulfur emissions from the CEDS anthropogenic emissions data. We chose to average over the period 2000-2015 as the different models studied here all use slightly different emissions years. We have now included a reference to Hoesly et al. in the figure caption.

*4. Table 1: The info for the indirect effects for HadGEM3 and IPSL seems to contradict the equivalent Table*

*1 of Liu et al., J. Clim. (2018), which describes the PDRMIP regional experiments. According to Liu et al., HadGEM3 includes both 1st and 2nd indirect effects for HadGEM3 sulfate, whilst IPSL includes 1st indirect effect only, which differ from the descriptions given here. Please double check the info.*

We thank the reviewer for pointing this out and have corrected this error in the manuscript.

*5. The present study uses 6 PDRMIP models (L142 and Table 1), but Liu et al. (2018) describes 7 PDRMIP models that ran the regional aerosol experiments. Why was CESM1-CAM4 not included in the present study, given that this model had apparently also run the PDRMIP regional experiments?*

We chose not to include the CESM1-CAM4 in our study because output from its PDRMIP BC10xASIA run was not available through NIRD. While it would be nice to include, we don't think its additional results would have significant bearing on the conclusions presented here, anyway.

*6. Methods and Results sections: On that point, the Liu et al. (2018) is not properly credited in this manuscript. The Liu et al. (2018) paper describes and presents initial analysis of the precipitation response to the PDRMIP regional experiments, including a brief discussion of the Asian monsoon response. This by no means detracts from the present study, which provides a much more in depth analysis of the Indian monsoon response in these PDRMIP experiments, however Liu et al. needs to be appropriately cited. Currently the only place I can find it referenced is in L223 where it is incorrectly referenced with regard to the global PDRMIP experiments, when in actual fact this paper analysed the regional experiments, and was the first to do so.*

We thank the reviewer for making this important point. We have now updated the citation in two places. We have replaced our initial description with: "The first regional analysis of the PDRMIP experiments by Liu et al. (2018) found also a weak precipitation response to BC changes, attributed to insignificant circulation changes relative to those induced by the sulfur experiments". We have also included a brief parantetical in the conclusions section.

*7. L246-247: I think this should say Figure S6 not S7.*

The reviewer is correct and we have now fixed this error.

*8. L282: "Almost all models" - there's only three models, so maybe just say "2 out of 3", otherwise it sounds more confident than it really is*

We thank the reviewer for pointing this out, and we note that the wording of our sentence was unclear. We meant that almost all scenarios across the multi-model ensemble show this increasing trend in precipitation, with the exception of just one scenario from one model. We have now corrected this sentence to say: "All scenarios across the multi-model ensemble (with the exception of CESM's CHN 20% SO2 scenario) show an increase in summer precipitation in India when SO2 emissions in China and/or India are reduced."

*9. Section 3.3/3.4: The authors could consider also referencing Shawki et al., JGRA (2018) in the discussion here, which found the same response of increased Indian monsoon precip in response to reducing Chinese SO2 emissions, using HadGEM3 (precursor to UKESM1), and attributed this to the increased land-sea temperature contrast. This supports your results here, and it could be good to note the consistency with this previous study.*

We thank the reviewer for making this point and have now added a sentence in section 3.4 addressing this point: "A similar analysis by Shawki et al. (2018) also found that reduced Chinese SO2 emissions strengthened the land-sea temperature contrast and consequently precipitation over India."

*10. L298-299: "For all reduced BC scenarios, the changes in India's precipitation are generally small (~5% locally) and not statistically significant at a 90% confidence level". Looking at Figure S8, this statement doesn't seem to be true for UKESM1.*

The reviewer is correct and we have added a short parenthetical to address this point.

*11. L361-362: Again Liu et al. (2018) should also be referenced here, as it previously showed this for the regional Asian BC PDRMIP experiment as well.*

We have now updated the sentence to include a reference to Liu et al.

---

## Referee Comment (RC2) · Anonymous Referee #2 · 29 Jan 2021

The authors analyzed South Asian mean precipitation response to BC and SO2 changes in multiple models. Overall this is a good study. I have a handful of comments, most of which are fairly minor. I apologize but I was not able to read the author response to the first review, which I notice was recently posted, so it is possible that some of these have already been addressed. Specific comments:

Line 67: needs citations

Line 68-74. Results (Fig S1 and Fig 1) should be in the results sections, not in the introduction section

Line 78-80. I think Westervelt et al. (2020) should be cited here, which looked at the relative contribution of aerosol vs greenhouse gas to the South Asian monsoon.

[Figure]

Westervelt, D. M., You, Y., Li, X., Ting,M., Lee, D. E., & Ming, Y. (2020).Relative importance of greenhouse gases, sulfate, organic carbon, and black carbon aerosol for South Asian monsoon rainfall changes. Geophysical Research Letters, 47, e2020GL0883 63

Line 105-107. A bit of a run on sentence with the multiple "ands" and the i.e. Consider rewriting.

Line 119-120. What is meant by "in more remote regions"? The US, or Europe? If so, a citation is needed to back up the claim that those regions emissions are less likely to change in a major way.

Line 123-124. About China's emissions likely decreasing and the Rao paper citation. Is this based on a certain SSP scenario? Do they all have Chinese emissions decreasing in the near term?

General introduction: In general the references in the introduction are a bit dated. There has been a lot of work on aerosol-climate in India since e.g. Ramanathan et al. 2005. Suggest a more thorough literature analysis.

Line 175. Conley et al 2018 is an incorrect citation here. That paper is about temperature. I believe the authors are referring to this paper instead:

Westervelt, D. M., A. J. Conley, A. M. Fiore,J.-F. Lamarque, D. Shindell, M. Previdi,G. Faluvegi, G. Correa, and L. W. Horowitz(2017), Multimodel precipitation responses to removal of U.S. sulfur dioxide emissions, J. Geophys. Res.Atmos., 122, 5024–5038, doi:10.1002/2017JD026756

Line 186-187. 40 years is rather short for precipitation responses in a coupled climate model, so this is an important detail that requires some additional explanation.

Line 227-228: What about the different aerosol schemes makes the ensemble spread so large? This is briefly mentioned and then abandoned immediately in favor of a cloud formation explanation.

[Figure]

Line 230-234: citations needed

Line 234: Is the semi-direct effect of BC included in every model?

Fig 2: For BC and sulfate, something odd is going on in the Had-GEM3 model. How much can we trust that model's result?

Line 282: Almost all models just means 2 of 3, right?

Line 284: The use of "regulations" here is odd and brings in an unjustified policy aspect. Suggest changing to "reductions"

Line 336. Work has already been done on comparison of sulfur dioxide vs greenhouse gases on the south Asian monsoon.

Conclusions/general comment: if even a tenfold increase in BC in ∼6 models still gives unclear and confusing results on the impact of BC on South Asian monsoon, is there really any hope to better understand the effect?

Line 382: This could also be a disbenefit, especially if aerosols impact extreme precipitation, in causing more flooding etc. There has been some work on aerosols and extreme precip that is probably worth citing.

Fig. 4: Between the colors, country borders, climatology, statistical significance stippling, and wind barbs, it is tough to decipher all of the layers of this figure. Perhaps wind speed and direction could be its own figure.
* * *

---

## Author Comment (AC2) · 3 Feb 2021

**Title: Sensitivity of modeled Indian Monsoon to Chinese and Indian aerosol emissions**

**_Response to Review #2_**

_We would like to thank the editor and reviewers for the helpful comments and for the thorough review of our paper._

We thank the reviewer for the careful review of our paper. We have addressed a number of questions and concerns from Reviewer #2. Detailed responses to the reviewer are given below. All of the changes in the manuscript are indicated in red in the Additional Material file.

**Response to Anonymous Reviewers**

**Reviewer: 2**

*Reviewer comments to the author:*

*The authors analyzed South Asian mean precipitation response to BC and SO2 changes in multiple models. Overall this is a good study. I have a handful of comments, most of which are fairly minor.*

We thank the reviewer for the helpful comments. We have addressed the comments and corrections suggested by the reviewer here, with changes indicated in red in the revised manuscript.

*Line 67: needs citations*

We have now included references to the relevant literature.

*Line 68-74. Results (Fig S1 and Fig 1) should be in the results sections, not in the introduction section.*

We feel that it's more appropriate to have these figures referenced in the introduction section as the figures are more relevant as background motivation to the study rather than part of the study itself.

*Line 78-80. I think Westervelt et al. (2020) should be cited here, which looked at the relative contribution of aerosol vs greenhouse gas to the South Asian monsoon.*

We have now included this reference here.

*Line 105-107. A bit of a run on sentence with the multiple "ands" and the i.e. Consider rewriting.*

We agree with the reviewer and have revised the sentence to the new form: "The purpose of the PDRIMP experiments here is to determine if a precipitation signal in the South Asian summer monsoon can be detected in scenarios with large emissions perturbations of sulfur and black carbon."

*Line 119-120. What is meant by "in more remote regions"? The US, or Europe? If so, a citation is needed to back up the claim that those regions emissions are less likely to change in a major way.*

We thank the reviewer for pointing out this ambiguity. By "more remote regions" we meant emissions from regions outside of India, like China, and this sentence has now been adjusted to reflect that.

*Line 123-124. About China's emissions likely decreasing and the Rao paper citation. Is this based on a certain SSP scenario? Do they all have Chinese emissions decreasing in the near term?*

We thank the reviewer for this question. Rao et al. specifically find that Asian emissions of $SO_2$ and $NO_x$ decrease in all five SSP scenarios in the near term. Additionally, they find a decrease in ozone concentrations between 5 and 14% in China by 2050.

*General introduction: In general the references in the introduction are a bit dated. There has been a lot of work on aerosol-climate in India since e.g. Ramanathan et al. 2005. Suggest a more thorough literature analysis.*

We thank the reviewer for making this point and have now included references to more recent

literature in the introduction section.

*Line 175. Conley et al 2018 is an incorrect citation here. That paper is about temperature. I believe the authors are referring to this paper instead: Westervelt, D. M., A. J. Conley, A. M. Fiore,J.-F. Lamarque, D. Shindell, M. Previdi,G. Faluvegi, G. Correa, and L. W. Horowitz(2017), Multimodel precipitation responses to removal of U.S. sulfur dioxide emissions, J. Geophys. Res.Atmos., 122, 5024–5038, doi:10.1002/2017JD026756*

We thank the reviewer for noticing this error and have fixed the reference in the manuscript.

*Line 186-187. 40 years is rather short for precipitation responses in a coupled climate model, so this is an important detail that requires some additional explanation.*

We agree with the reviewer that it is true that 40 years is a short duration to detect precipitation responses at a fine temporal resolution. We have now clarified the sentence to note that 40 years of data is sufficient to detect signal in the seasonally aggregated precipitation statistics.

*Line 227-228: What about the different aerosol schemes makes the ensemble spread so large? This is briefly mentioned and then abandoned immediately in favor of a cloud formation explanation.*

We have updated this sentence to include more information and a reference to a relevant paper analyzing the effect of aerosol scheme on cloud response: "It is likely that different aerosol schemes across models (Table 1) may be implicated as one of the dominant sources of the large ensemble spread by altering simulated clouds radiative properties and lifetimes, as has been shown in previous studies testing different aerosol schemes in the same coupled climate model (Nazarenko et al., 2017). Additionally, both the boundary layer scheme and modelling impacts of absorbing aerosols on cloud formation could also play important roles."

*Line 230-234: citations needed.*

We have now included a reference to Koch, D. and Del Genio, A.D., 2010: Black carbon semi-direct effects on cloud cover: review and synthesis. Atmos. Chem. Phys., 10, 7685-7696.

*Line 234: Is the semi-direct effect of BC included in every model?*

The semi-direct effect of BC is included in some form in every model studied here. However, differences in parameterizations of chemical, physical and dynamical processes can lead to a wide range of semi-direct responses across models (Stjern et al. 2017).

*Fig 2: For BC and sulfate, something odd is going on in the Had-GEM3 model. How much can we trust that model's result?*

The reason for the strange HadGEM3 results in Figure 2 is likely because precipitation in the CTRL simulation for the model is very low over most of India (see Figure S3 for reference), which means small changes in precipitation in the BC and sulfate experiments could result in large percent changes in precipitation. We have now included this information in the main text: "The weak precipitation over India in HadGEM3 in the CTRL simulation (Figure S3) also likely explains the large percent changes indicated in the BC and sulfate experiments."

*Line 282: Almost all models just means 2 of 3, right?*

We thank the reviewer for pointing this out, and we note that the wording of our sentence was unclear. We meant that almost all scenarios across the multi-model ensemble show this increasing trend in precipitation, with the exception of just one scenario from one model. We have now corrected this sentence to say: "All scenarios across the multi-model ensemble (with the exception of CESM's CHN 20% SO2 scenario) show an increase in summer precipitation in India when SO2 emissions in China and/or India are reduced."

*Line 284: The use of "regulations" here is odd and brings in an unjustified policy aspect. Suggest changing to "reductions"*
We agree with the reviewer and have revised the sentence accordingly.

*Line 336. Work has already been done on comparison of sulfur dioxide vs greenhouse gases on the south Asian monsoon.*
We thank the reviewer for this point and have revised the sentence appropriately: "Changing circulation patterns are suggested as a consequence of changes in CO2 as well, and potential nonlinear effects of sulfur and greenhouse emissions on monsoon precipitation highlight an important challenge in predicting future changes to the South Asian summer monsoon."

*Conclusions/general comment: if even a tenfold increase in BC in ~6 models still gives unclear and confusing results on the impact of BC on South Asian monsoon, is there really any hope to better understand the effect?*
It is true that it is at present difficult to understand the role of BC on the South Asian summer monsoon given the wide range of results from the multimodel ensembles. That being said, computational improvements in the future should allow for higher resolution simulations of vertical BC profiles, which should reduce one of the large points of uncertainty in these simulations. Additionally, improvements in parameterizations of sub-grid processes for cloud formation may play an important role, as well. Also, it should be noted that since PDRMIP there have been significant changes in our understanding of BC and that this is an evolving field (i.e. the role of coating affecting its radiative properties).

*Line 382: This could also be a disbenefit, especially if aerosols impact extreme precipitation, in causing more flooding etc. There has been some work on aerosols and extreme precip that is probably worth citing.*
We thank the reviewer for making this important point and have included an additional sentence to reflect this: "It is, however, important to bear in mind that SO2 emissions reductions could also increase flooding and extreme precipitation generally (Sillmann et al., 2019)."

*Fig. 4: Between the colors, country borders, climatology, statistical significance stippling, and wind barbs, it is tough to decipher all of the layers of this figure. Perhaps wind speed and direction could be its own figure.*
We agree with the reviewer that the figure was too dense before, and have moved the quiver plots to the SI as Figure S10.